# Comparative Analysis of Physical and Chemical Properties of Differently Obtained Zn—Methionine Chelate with Proved Antibiofilm Properties (Part II)

**DOI:** 10.3390/pharmaceutics15020590

**Published:** 2023-02-09

**Authors:** Alla V. Marukhlenko, Vladimir N. Tumasov, Leonid A. Butusov, Georgy A. Shandryuk, Mariya A. Morozova

**Affiliations:** 1Department of Pharmaceutical and Toxicological Chemistry, Peoples Friendship University of Russia (RUDN University), 6 Miklukho-Maklaya Street, 117198 Moscow, Russia; 2Institute of Innovative Engineering Technologies, Peoples Friendship University of Russia (RUDN University), 6, Miklukho-Maklaya st., 117198 Moscow, Russia; 3Russian Academy of Sciences A.V. Topchiev Institute of Petrochemical Synthesis, 29/2, Leninsky Prospect, 119071 Moscow, Russia

**Keywords:** zinc, methionine, metal organic compounds, nanoparticles, α- and β-polymorphic forms, optical rotation, geometric isomers

## Abstract

The previously demonstrated activity of aqueous solutions of methionine and zinc salts against biofilms of uropathogenic bacteria prompted us to investigate the structure and properties of zinc methionine complex obtained from such solutions. The paper presents the analysis results of zinc coordination complexes with methionine obtained by synthesis (0.034 mol of L-methionine, 0.034 mol of NaOH, 40 mL of H_2_O, 0.017 mol ZnSO_4_, 60 °C) and simple crystallization from water solution (25 mL of a solution containing 134 mmol/L L-methionine, 67 mmol/L ZnSO_4_, pH = 5.74, I = 0.37 mmol/L, crystallization at room temperature during more than two weeks). IR spectral analysis and X-ray diffraction showed the structural similarity of the substances to each other, in agreement with the data described in the literature. DSC confirmed the formation of a thermally stable (in the range from −30 °C to 180 °C) chelate compound in both cases and indicated the possible retention of the polymorphic two-dimensional structure inherent in L-methionine with the temperature of phase transition 320 K. The crystallized complex had better solubility in water (100 to 1000 mL per 1.0 g) contra the synthesized analog, which was practically insoluble (more than 10 000 mL per 1.0 g). The results of the solubility assessment, supplemented by the results of the dispersion analysis of solutions by the dynamic light scattering method indicated the formation of zinc-containing nanoparticles (80 nm) in a saturated water solution of a crystallized substance, suggesting the crystallized substance may have higher bioavailability. We predicted a possibility of the equivalent existence of optically active cis and trans isomers in methionine-zinc solutions by the close values of formation enthalpy (−655 kJ/mol and −657 kJ/mol for cis and trans forms, respectively) and also illustrated by the polarimetry measurement results (∆α = 0.4°, pH = 5.74, C(Met) = 134 mmol/L; the concentration of metal ion gradually increased from 0 to 134 mmol/L). The obtained results allowed us to conclude that the compound isolated from the solution is a zinc-methionine chelate with the presence of sulfate groups and underline the role of the synthesis route for the biopharmaceutical characteristics of the resulting substance. We provided some quality indicators that it may be possible to include in the pharmacopeia monographs.

## 1. Introduction

Antibiotic resistance and biofilms are becoming increasingly common problems in today’s medical world [1]. Bacteria and other microorganisms have evolved to survive and even thrive in the presence of antibiotics. This has led to an increasing number of infections that are resistant to traditional treatments, resulting in the need for more aggressive and often more costly treatments [2]. To address the problem of antibiotic resistance and biofilm formation, researchers are investigating new treatments and strategies [3]. For example, nanotechnology offers alternative sources to antibiotics: specific properties of nanoparticles, such as size, shape, and surface chemistry can be used to create an environment that is inhospitable to the growth of biofilms, thus preventing them from forming and protecting the host from infection [4]. Metal chelation is also a promising strategy for the development of new compounds to treat antibiotic-resistant infections. It has the potential to create novel compounds with increased efficacy and reduced side effects [5,6,7,8]. However, further research is needed to fully understand the mechanisms of metal chelation and to optimize the design of chelators for maximum efficacy. The selection of a chelating metal is primarily based on its toxicity, while factors such as activity, solubility, lipophilicity of ligands are also considered [9]. In this regard, zinc compounds are proving to be promising due to their essential biogenic element status and comparatively low toxicity [10].

Examining the antibacterial characteristics of zinc containing compounds, two contrasting opinions can be recognized within scientific reviews and papers. There is proof of the essentiality of zinc not only for eukaryote metabolism but also for bacteria [11]. Several studies have demonstrated that in the presence of zinc ions, biofilm growth increases greatly resulting in a stable antibiotic resistant form [12]. This is attributed to the existing complex systems of zinc homeostasis regulation, by both the bacterial and host cells [13]. These works mainly investigated the effect of inorganic zinc salts, that are fully ionized under physiological conditions or in culture fluid. However, conflicting results showed nanoparticles of zinc or its oxide exhibiting distinct antibiofilm properties [14]. Researchers proposed multiple mechanisms for the damage of cells induced by ZnO nanoparticles including direct interaction with a bacterial cell, damage to cell integrity, the liberation of ROS species, damage of lipids, proteins, carbohydrates and DNA by oxidative stress [15]. Although the actual mechanism is not fully understood and is still controversial, the antimicrobial properties of ZnO nanoparticles have made them a potential candidate in a wide range of biomedical applications [16]. That is why increasing attention has been drawn to developing novel synthesis routes to generate Zn organic nanoparticles [17,18,19]. The experimental conditions usually not only determine the size of obtained crystals but affect their porosity, morphology, and properties [20]. Thus, it is of great importance to properly choose a synthesis method that controls the physiochemical and biopharmaceutical characteristics of the acquired products.

In one of our past works, we assessed the toxicity and predicted the biological activity of aqueous solutions of zinc salts with methionine: the results of microbiological experiments proved that a zinc-containing supplement aimed at eliminating the deficiency of this element is safe concerning the natural intestinal microflora, restraining the growth of pathogenic strains [21]. Obviously, in the analyzed aqueous solutions, the active component was a chelate compound of an amino acid and a metal, not excluding the probability of their existence in nanoparticle form. The possibility of chelate formation in neutral aqueous solutions was simply worked out knowing the value of stability constant, pH, and molar concentrations of the starting components. Actually, the crystal substance of zinc-methionine (Zn(Met)_2_) can be easily obtained by a chelation reaction of methionine in an alkaline medium. This is done by heating the amino acid in the alkaline medium with a zinc salt [22]. This reaction leads to a chelate complex formation, with zinc-methionine, an almost white powder with well-studied and described crystal structure, being the main product [18,23]. However, its exceptionally low water solubility imposes many limitations for subsequent applications. For example, the antibacterial properties of this compound have been proven, but only in an experiment using DMSO as a solvent, which does not exclude the possibility of a synergistic reaction. Thus, the authors proposed the further use of this complex exclusively for the treatment of infectious diseases of a dermatological nature [24]. Assuming all of the above, we decided to obtain a water-soluble zinc methionine complex, applicable for biological testing, by crystallizing it from a solution containing two pharmacopoeial substances—L-methionine and zinc sulfate. Characterization of structure and possible composition of the complexes obtained by crystallization at room temperature, in comparison with those compounds obtained in classical synthesis from an alkaline solution of methionine, seemed to be an interesting scientific task.

## 2. Materials and Methods

### 2.1. Chemicals

L-Methionine (pure, pharma grade, AppliChem, Barcelona, Spain), zinc sulfate monohydrate (99.0%, Acros Organics, Barcelona, Spain) and all other reagents and chemicals (sodium hydroxide, ethanol) used were of analytical grade. For dilution, highly purified water was used, obtained using the Milli-Q^®^ purification system (Merck, Darmstadt, Germany).

### 2.2. Synthesized Complex Zn(Met)_2_

The method that was used for the synthesis of zinc methionine is not difficult in technical execution and is characterized by high product yield. The synthesis method was described in our works earlier [18,24]. Sodium hydroxide (0.034 mol) was added to the suspension of L-Methionine (0.034 mol) in 40 mL of water, stirred for 10 min until the L-Methionine was completely dissolved and then the mixture was heated to 60 °C. Zinc sulfate monohydrate (0.017 mol) was added to the homogeneous solution in portions and kept under stirring for 20 min at a temperature of 60 °C. Next, the reaction mixture was cooled to 10 °C, the precipitated crystals were filtered and washed with water until there was a negative reaction to sulfates (reaction with barium chloride) and to free methionine (ninhydrin test). The residue was additionally washed with ethyl alcohol and dried to a constant mass at 105 °C in a dry heat cabinet with forced convection (Binder RF 53, Germany) [25]. The appearance of the studied compounds was determined using 12 MP smartphone camera with 2× optical zoom out.

### 2.3. Isolation of the Zn(Met)_2_(SO_4_)_x_·nH_2_O Crystals

Crystallized zinc methionine sulfate was obtained by the complexation reaction when mixing aqueous solutions of ZnSO_4_ and Met. The equilibrium concentration of the product was estimated at different metal-ligand ratios based on the data of stability constants [26]. An excessive amount of ligand (Zn:Met = 1:20) shifts the complexation reaction toward product Zn(Met)_2_. However, in this case, the residual amino acid remains in the system when the solvent evaporates, and the precipitate crystallizes. In further experimental work, the metal and a ligand were used in the stoichiometric ratio Zn: Met = 1:2 to exclude the described effect. For example, the concentration of the complexing agent and ligand was 67 mmol/L and 134 mmol/L for Zn^2+^ and Met, respectively. The obtained solution was adjusted to a pH value of the isoelectric point of Met (pI = 5.74) with 2.12 mol/L NaOH which excluded Zn(OH)_2_ precipitation. The obtained solution was evaporated at room temperature to a constant mass of dry residue. Probable composition of the obtained crystals: Zn(Met)_2_(SO_4_)_x_·nH_2_O.

### 2.4. Solubility

The water solubility of the samples was evaluated in accordance with the gradation of the European Pharmacopoeia based on a ratio of the approximate solvent volume per 1.0 g of the substance [27].

### 2.5. Dynamic Light Scattering (DLS)

A Zetasizer Nano ZSP (Malvern Panalytical, Worcestershire, UK) based on dynamic light scattering was used to measure the size of nanoparticles in the saturated solutions obtained in the solubility test. Disposable polystyrene cuvettes, filled with 1 mL of sample, were used. For each size determination, three replicate measurements were performed, and the average size value was calculated. Each measurement consisted of 12 runs. The refractive index value was 1.3365.

### 2.6. Fourier Transform Infrared (FT-IR) Spectroscopy

To analyze the samples in the spectral range from 4000 to 750 cm^−1^, an IR Fourier spectrophotometer Agilent Cary 630 (Agilent, Santa Clara, CA, USA) with a diamond ATR accessory was used. The resolution is less than 2 cm^−1^, the correctness of the wavenumber is 0.05 cm^−1^, and the reproducibility of the wavenumber is 0.005 cm^−1^. The thickness of the absorbing layer is 1.5 nm (the clamping device guarantees the setting of optimal and reproducible pressure). To confirm the presence of characteristic bonds in the synthesized and crystallized complexes, L-Methionine and sodium methionine were selected as controls and investigated under the same measurement conditions. The standard Agilent MicroLab Expert software was used to control the device, measure the data, and evaluate the quality of the obtained spectra; in this case, the FTIR spectra were visualized in the wave number, cm^−1^—transmission, % coordinates.

### 2.7. X-ray Fluorescence Analysis (XRF)

The presence of zinc in the obtained samples was controlled by the XRF method. An EDX-7000 Shimadzu energy dispersive X-ray fluorescence spectrometer (Shimadzu, Kyoto, Japan) equipped with the PCEDX-Navi software (Shimadzu, Japan) package was used to carry out the non-destructive elemental composition study of powder samples. The range of elements was measured—11 Na—92 U; X-ray generator—a tube with a Rh-anode, air-cooled; voltage 4–50 kV, current 1–1000 μA; irradiated area—a circle of 10 mm in diameter; silicon drift detector (SDD), counting method—a digital counting filter; automatic change of filters; chamber size 300 mm × 275 mm × 100 mm. The X-ray fluorescence spectrum for each measurement was recorded at the same device settings: mylar film, collimator—10 mm, exposure time—100 s, atmosphere—air; the number of repeated measurements for one sample n = 3. The fluorescence signal intensity was measured at K_α_ and K_β_ zinc lines (8.632 and 9.572 keV, respectively). The results obtained using the XRF method are presented in values of fluorescent intensity expressed in cps/µA.

### 2.8. Thermogravimetric Analysis (TGA)

Thermogravimetric analysis of synthesized Zn(Met)_2_ was carried out using a Thermal Analysis System TGA/DSC ^3+^ instrument (Mettler Toledo, Nänikon, Switzerland). Details: sample mass—8–10 mg, the crucible—Al_2_O_3_, V = 150 µL. The compound was heated in a static atmosphere of air from 298—1273 K, with a heating rate of 10 K/min. The results of thermogravimetric analysis are presented as a function of the change in the mass of the sample with temperature.

### 2.9. Complexometric Titration

Quantitative assessment of the zinc content in the synthesized complex Zn(Met)_2_ was carried out by complexometric titration following the methodology of the European Pharmacopoeia: 0.200 g of the synthesized complex was dissolved in water with the addition of 5 mL of ammonium chloride buffer solution (pH 10.0) and about 50 mg of mordant black 11 triturate and then titrated with 0.1 M sodium edetate until the color changed from violet to full blue (n = 3) [28,29].

### 2.10. Differential Scanning Calorimetry (DSC)

Differential scanning calorimetry of synthesized Zn(Met)_2_ and crystallized Zn(Met)_2_(SO_4_)_x_·nH_2_O complexes were carried out using Thermal Analysis System DSC ^3+^ (Mettler Toledo, Switzerland) with a liquid nitrogen cooling system, sensor—HSS9^+^. The substance L-Methionine was used as a control. The measurements were carried out sequentially in the heating—cooling—reheating mode in the temperature range (−223)—463 K, in an atmosphere of N_2_ (50.0 mL/min), in an aluminum crucible (40 µL).

### 2.11. Optical Activity

The optical activity was investigated for both types of zinc chelate complexes with methionine. Two solutions were prepared for the synthesized zinc methionine complex Zn(Met)_2_: 14 mmol/L solution in HCl (pH = −1.35) and 14 mmol/L solution in NaOH (pH = 12.95). 134 mmol/L methionine solutions prepared under the same conditions were used as a control, and the pH of the solutions was −1.35 and 12.95, respectively.

To study the optical activity of aqueous solutions of zinc methionine (Zn^2+^: Met = 1:20, 1:10, 1:5, 1:2, 1:1) samples were prepared under conditions: C(Met) = const = 134 mmol/L, zinc concentration in solution gradually increased; pH = pI(Met) = 5.74. Aqueous solution of Met (134 mmol/L) prepared under the same conditions was used as a control. For ionic strength adjustment an appropriate amount of NaCl from stock solution was added to the control methionine solution.

The optical activity was determined using the Atago POL-1/2 polarimeter (Atago Co., Ltd., Fukui, Japan), in a 100 mm cell, λ = 589.3 nm, the measurement accuracy ±0.002°, the resolution 0.0001°. The electronic Peltier module was used for setting the required temperature (T = 20 ± 0.5 °C). The angle of optical rotation was measured for 1 min in 5 repetitions.

### 2.12. X-ray Powder Diffraction (XRD)

XRD patterns were obtained with a Bruker d8 advance diffractometer using 1.541 Å (Cu-K_α_) radiation at room temperature in the 2θ range of 5–80°. All samples were normalized by weight. For simulation of XRD patterns VESTA software was used. Crystallographic data for L-methionine, and complexes was obtained from Cambridge crystal structure database.

### 2.13. Molecular Modeling and Data Processing

Calculations and all mathematical transformations of the spectral data were performed using the OriginPro 2017 software (OriginLab, Northampton, MA, USA). MolView and WebMo programs were used for modelling molecular structures and for enthalpy prediction, respectively [30,31].

## 3. Results and Discussion

### 3.1. Obtained Complexes

Using the above methods, two compounds Zn(Met)_2_ and Zn(Met)_2_(SO_4_)_x_·nH_2_O were obtained. The yield of the synthesized product Zn(Met)_2_ was 70.3%, the yield of the product after the crystallization process was not evaluated. The appearance of the obtained chelated compounds is shown in Figure 1: synthesized zinc methionine complex Zn(Met)_2_ is white crystalline powder, crystallized zinc methionine complex Zn(Met)_2_(SO_4_)_x_·nH_2_O represents colorless or white crystal needles.

### 3.2. Solubility

It was established experimentally that the synthesized complex Zn(Met)_2_ is practically insoluble in water (more than 10,000 mL of solvent per 1.0 g of the substance), while the crystallized complex Zn(Met)_2_(SO_4_)_x_·nH_2_O is slightly soluble in water (100 to 1000 mL of solvent is consumed per 1.0 g of the substance). 

### 3.3. Dynamic Light Scattering (DLS) 

The solutions obtained in the solubility testing were investigated by the dynamic light scattering method to determine the hydrodynamic radius of the presenting particles. To characterize the dispersed systems, the function (scatter intensity fraction, %—d, nm) was used. The DLS technique allowed detection of particles from 340 to 712 nm with a maximum of 460 nm in the solution of synthesized Zn(Met)_2_. The solution of crystallized Zn(Met)_2_(SO_4_)_x_·nH_2_O shows a bimodal distribution in the region from 35 to 712 nm with peaks at 78.8 and 396 nm (see Figure 2). The picture of the volume distribution of particles completely repeated the result expressed in units of intensity.

Dissolution of crystallized zinc methionine in water leads to the presence of nanoparticles capable of overcoming cell membranes by various means of transportation due to size. The results of the solubility testing and dispersed composition of solutions suggest that crystallized Zn(Met)_2_(SO_4_)_x_·nH_2_O, in comparison with synthesized Zn(Met)_2_, may have higher pharmacokinetic parameters (absorption and distribution) and does not require additional delivery systems to target organs when administered orally [32,33]. Most likely, the improvement of solubility in water is due to the presence of sulfate anions in the composition of crystallized zinc methionine [34].

### 3.4. Fourier Transform Infrared (FT-IR) Spectroscopy 

According to literature data, complexes of divalent metal cations with methionine are characterized by O, N-type of binding without participation in the complexation of the methyl sulfide group [35]. A comparison of the obtained IR spectra of L-methionine, sodium methionine, and synthesized complex Zn(Met)_2_ confirms the proposed structure (see Figure 3). On the IR spectrum of the synthesized complex, two bands of a coordinated amino group (3339, 3252 cm^−1^) are observed with a split of Δv ≈ 87 cm^−1^; the latter is consistent with the ideas of a donor-acceptor bond between a metal and a nitrogen atom of the amino group [24]. On the spectrum of synthesized zinc methioninate, as well as sodium methioninate, vibration of the CH_3_S- group is preserved at 1327 cm^−1^, i.e., the sulfur atom does not participate in the coordination of zinc. In the spectra of NaMet and Zn(Met)_2_, there is no absorption band at 1503 cm^−1^, which indicates the participation of the carboxyl group in the formation of a covalent polar bond between O and Zn. The absence of significant absorption bands above 3450 cm^−1^ indicates the absence of an O-H bond, and, consequently, molecules of both crystallized and adsorbed water.

The IR spectrum of L-methionine, sodium methioninate, and the crystallized zinc methioninate complex Zn(Met)_2_(SO_4_)_x_·nH_2_O are shown in Figure 4. The IR spectrum of the crystallized complex also retains the CH_3_S- group oscillation-band at 1327 cm^−1^, besides there is no absorption band at 1503 cm^−1^, which corresponds to the carboxyl group (-COOH). However, two bands of the coordinated amino group (3339, 3252 cm^−1^) are very weakly expressed on the IR spectrum of the crystallized complex, which is more noticeable on the difference spectrum (Figure 4, insert). Thus, the formation of the covalent-polar bond O-Zn between the metal and the amino acid residue is strongly pronounced on the obtained IR spectrum and the fluctuations of the donor-acceptor bond N→Zn are much weaker. The absorption bands at 3130 and 2100 cm^−1^ are present on the L-methionine spectrum which indicates protonation of the amino group. Similar fluctuations were also observed in the IR spectrum of crystallized zinc methioninate Zn(Met)_2_(SO_4_)_x_·nH_2_O that may reveal the presence of free methionine in the sample, i.e., the formation of zinc chelate complex in the stoichiometric ratio Zn: Met ≠ 1:2. In addition, the absence of significant absorption bands in the frequency range above 3450 cm^−1^ means the absence of oscillations of the O-H bond, and, consequently, crystallization water molecules. Thus, the composition of the crystallized complex can be described by an adjusted formula without a crystalline hydrate water—Zn(Met)_2_(SO_4_)_x_.

### 3.5. X-ray Fluorescence Analysis (XRF)

X-ray fluorescence analysis was used to demonstrate the zinc content in the synthesized and crystallized zinc methioninate complexes. The presence of Zn in the studied samples was proved by a characteristic peak in the XRF spectra at the zinc K_α_ and K_β_ lines (8.632 and 9.572 keV, respectively) (see Figure 5).

### 3.6. Thermogravimetric Analysis (TGA)

The analysis of TGA/DTA results shows a decrease in the mass of the sample Zn(Met)_2_ by 0.35% at 488.69 K; the mass loss is due to an unidentified component, but probably not water [24]. The decomposition of synthesized Zn(Met)_2_ was recorded in the temperature range from 581.4 to 1073 K (Figure 6). Two different mass losses were observed in the TGA/DTA diagram with maximum decomposition rates at 598.44 and 820.04 K. Step one and step two mass losses were about 63.10% and 12.62%, respectively. The first step occurred in the range of 573 to 673 K and may be related to losing the carboxyl group into CO_2_. The second step from 783 to 1073 K is due to the loss of NH_2_ group and sulfur moiety in the complex [20,24]. According to the TGA analysis results, the zinc content in the synthesized chelate complex is 19.23%, while the calculated Zn content for the complex described by formula Zn(Met)_2_ is 18.07%.

### 3.7. Complexometric Titration

Complexometric titration of the synthesized zinc methionine Zn(Met)_2_ after its dissolution in an ammonium chloride buffer solution (pH 10) showed the average zinc content is 18.23% (n = 3, RSD = 1.18%).

The results of complexometric titration and thermogravimetric analysis suggest a breach of the stoichiometric ratio of zinc to methionine in the synthesized complex 1:2. The results obtained correlate with the literature data representing the structure of zinc methionine in such a way that one zinc atom is bound to more than two methionine molecules, but each methionine molecule is bound to one zinc atom [23].

### 3.8. Differential Scanning Calorimetry

DSC thermogram for L-Methionine powder is shown in Figure 7 (black). The L-Methionine powder shows a typical endothermic transformation upon heating at a temperature range from 283 to 321 K with a peak at 309 K. It is known that the phenomenon of polymorphism is typical for L- and DL-Methionine. In some works, three possible crystal forms of methionine (α, β and γ), as well as the processes of polymorphic transitions between them are described [36,37,38]. In both α- and β- methionine, the atoms C1, C2, C3, C4 and S form an almost planar zigzag chain. For α-methionine, C5 by the free rotation of the bond C5S lies out of this plane, whereas for β -methionine it is in the plane [37]. The free rotation of the terminal CH_3_-S group causes two different ways of achieving a stable balanced configuration of 2D hydrogen-bonded bilayers interconnected by weak Van der Waals interactions [36,37,38]. It is known that α- and β-polymorphic forms have a thermodynamic transition point between 274 and 323 K [39,40]. In this case, the crystals transform layer wise, without complete delamination or deterioration, and with a transition front that spreads perpendicular to the layers [37].

According to our measurements, we observed the thermodynamic transition point β→α forms of the L-Met that corresponds to the described data. The transition is reversible and reproducible over repeated cycles of the same sample: after cooling and reheating the preservation of the endothermic peak at the same temperatures was noted.

DSC thermograms of synthesized and crystallized zinc methionine are also shown in Figure 7 (red and blue, respectively). The thermogram for synthesized Zn(Met)_2_ crystals show a shift of the thermodynamic transition point (β→α): the endothermic peak is recorded in the temperature range from 316 to 328 K with a maximum at 320 K. The obtained results confirm the formation of the chelated compound of methionine, at the same time, indicating the possible preservation of polymorphism in the structure of the chelate complex Zn(Met)_2_. On the thermogram for crystallized Zn(Met)_2_(SO_4_)_x_, two endothermic peaks in the temperature range from 283 to 327 K with maxima at 309 K and 318 K were noted. This indicates the mixed nature of the compound. The maximum at 318 K proves the complexation reaction between Zn^2+^ and Met with the formation of a chelated zinc compound in one of possible polymorphic forms. The peak at 309 K confirms the presence of an excess of free L-Met in β form in Zn(Met)_2_(SO_4_)_x_ crystals after the complexation reaction and subsequent solvent removal. This is also confirmed by the preserved endothermic transition, like that observed in methionine at 394 K, but much more pronounced on the thermogram. Interesting is the appearance of a new polymorphic transition at 411 K, which is not characteristic of any of the other compounds—apparently, this transition is due to the formation of a chelate, including sulfate fragments.

### 3.9. Optical Activity

Studies of the optical activity of Zn(Met)_2_(SO_4_)_x_ aqueous solutions were carried out at pH = pI(Met) = 5.74. It is a difficult task to estimate the concentration of the final complex product in the obtained solution when mixing Zn^2+^ and Met solutions in different ratios. Therefore, the results of the polarimetric analysis are presented as a dependence of the optical rotation angle on the concentration of Zn^2+^ in the solution. The control solution of 6.7 mmol/L ZnSO_4_, as expected, did not show any optical activity (α = 0.00° ± 0.00, n = 5). When mixing solutions containing Zn^2+^ and Met in molar ratios 1:20, 1:10, 1:5, 1:2, 1:1, it was noted that the optical activity of the obtained solutions differed from the optical activity of the methionine solution (see Figure 8a). The formation of an inverse exponential relationship between the values of the optical activity and the concentration of zinc ions in solutions was detected: with an increase in the concentration of Zn^2+^, the angle of optical rotation of the plane polarized light decreased. The ionic strength of the solutions, as expected, slightly increased with increasing zinc concentration, from 0.1 mol/L in a solution of methionine with pH = 5.74 to 0.64 mol/L in a solution of zinc methionine with maximum metal content and adjusted pH. But this did not have a significant effect on the angle of rotation in solution (Figure 8b).

The phenomenon of the formation of a new chirality axis in chelated compounds of amino acids with metals (Ni, Co, Cu, Pt, etc.) due to the formation of Me-O and Me-N bonds is described in scientific periodicals: two amino acids anions coordinate to the metal by forming a trans or cis square-planar structure with two chelate rings [41,42,43]. The obtained values of the optical activity of the chelate complex of methionine confirm the described phenomenon: with an increase in Zn^2+^ concentration in the methionine solution the proportion of coordinated amino acid molecules increases and, accordingly, so does the number of new chirality centers formed. However, no changes in optical rotation are observed by mixing Zn^2+^ and Met solutions in molar ratios >1:1. This phenomenon can be explained by the formation of a limiting number of coordination bonds between a free amino acid and metal cations in a molar ratio of 1:1 with the formation of the maximum possible numbers of new chirality centers. The subsequent addition of an excessive amount of Zn^2+^ (>134 mmol/L) does not change the optical activity of the solution due to the absence of free ligands in it.

When trying to measure the optical activity of the same samples in a neutral medium (pH = 7.00), it was possible to obtain results only for a solution with a ratio Zn^2+^: Met—1:20. For all other combinations with a higher chelating agent content, starting with a solution of 1:10, amorphous precipitation of Zn(OH)_2_ was observed. The obtained results indicate a shift in the equilibrium towards the course of a competitive precipitation process in solutions (Figure 9).

The key factor is probably an increase in the equilibrium concentration of [Zn^2+^] at a constant concentration of the ligand [Met] and a constant pH value = 7.00, hence [Zn^2+^] × [OH^−^]^2^ > K_sp_ > β_2_. It is logical that in the case of complexation at an isoelectric point (pH =5.74), the reverse pattern is observed: K_sp_ < β_2_—firstly, due to a decrease in the concentration of hydroxide anions, and secondly, probably because the amino acid in the form of a zwitter-ion binds more easily to the metal.

We assume the possibility of the formation in the solution of two isomeric forms of the chelate complex Zn(Met)_2_(SO_4_)_x_—cis- and trans-, whose spatial structures, excluding water molecules and sulfate anions, are shown in Figure 10 [44]. The calculated values of the formation enthalpy may indicate the simultaneous existence in a solution of both forms of chelate complexes cis- and trans- Zn(Met)_2_(SO_4_)_x_ in an equivalent amount.

The synthesized zinc chelate complex with methionine Zn(Met)_2_ is practically insoluble in water, therefore its optical activity was studied in aqueous solutions at pH −1.35 and 12.95. As a control, 134 mmol/L L-methionine solutions prepared under similar conditions were used. Table 1 shows the obtained results of the optical activity of the studied solutions in units of specific rotation [α]^20^_D_.

Differences in the optical activity of the free amino acid and its chelate complex with zinc are demonstrated: the values of the specific rotation of Zn(Met)_2_ solutions are lower regardless of pH level. In addition, an alkaline solution of zinc methionine rotates the plane-polarized light to the left. The differences in the values of the specific rotation of the zinc methionine and the control confirm the fact that the chelate structure of the complex with a new optical center is formed during the synthesis process and is not destroyed under the action of a strong acidic or highly alkaline medium. It is interesting to note that when measuring the optical rotation of a sample obtained by mixing solutions of zinc sulfate and methionine in a ratio of 1 to 20, under conditions of pH = −1.35 and pH = 12.95, no significant difference was found between the measured optical rotation of the mixture and a pure amino acid. This allows us to conclude that there are no structures in the solution with additional chirality axes, i.e., the binding of a zinc atom and an amino acid molecule in an anionic or cationic form is not followed by the formation of a chelate structure.

It is interesting to note that a small fraction (30%) of particles about 70 nm in size was found in the solution obtained by mixing zinc sulfate and methionine in a ratio of 1 to 2. But a strong and stable signal was recorded in an alkaline solution of zinc methionine Zn(Met)_2_, indicating the presence of a fraction of nanoparticles about 50 nm in size. Thus, when zinc methionine is dissolved in an alkaline medium, the chelate structure of the compound is preserved with the formation of zinc-containing nanoparticles [18]. At the same time, the formation of the Zn(Met)_2_(SO_4_)_x_ complex in an alkaline medium by mixing solutions of methionine and zinc sulfate proceeds slightly.

### 3.10. X-ray Powder Diffraction

The XRD analysis of synthesized and crystallized zinc methionine complexes is shown in Figure 11. Recrystallized L-Methionine was used as a control. The crystals of L-methionine, in the form of thin elongated plates, were obtained by evaporation of water solutions at room temperature. The pattern for L-Met shows characteristic peaks in 2θ of 6.159°, 11.621°, 22.986°, 29.441° and 35.454° (see Figure 11). 

A few differences in the positions of characteristic peaks on the XRD spectra for zinc-chelated compounds relative to the spectrum of pure L-Methionine were found. The XRD pattern of a synthesized Zn(Met)_2_ has characteristic peaks in 2θ of 12.0128°, 17.677°, 20.138°, 21.858°, 25.028° and 26.241°. The crystal structure of the synthesized complex matches the nano Zn(II)-Meth 1:2 complex depicted in the Cambridge crystallographic data center (CCDC) with characteristic peaks in 2θ of 11.449°, 17.025°, and 20.369° [21]. At the same time, a crystallized Zn(Met)_2_(SO_4_)_x_ has characteristic peaks in 2θ of 11.979°, 17.778°, 23.341°, 25.275°, 27.2856°, 29.241°. The obtained pattern for the crystallized complex is similar to the synthesized one with possible differences due to the presence of sulfates in the structure.

## 4. Conclusions

As a result of the work carried out, we proved the structural similarity of zinc complex compounds with methionine obtained by two different methods—synthesis and crystallization from water solution. 

In addition to the structural similarity, confirmed by IR spectroscopy and X-ray diffraction, several important characteristics of the crystallized substance were revealed, which are essential from the point of view of the biopharmaceutical approach. For example, we have shown that the solubility of a crystallized substance in water is several orders of magnitude higher, which is undoubtedly a great advantage when creating dosage forms. In an aqueous solution of both substances, the formation of zinc-containing nanoparticles, which probably have an antibacterial effect, is observed. The phenomenon of crystal dimorphism inherent in L-methionine is preserved in both substances, while thermal analysis showed the stability of the compounds in the temperature range from −30 to 180 °C. It is promising to study the chiral properties of zinc and amino acid chelate compounds to separate geometric isomers and determine their biological effect.

The obtained results underline the role of the synthesis route for the biopharmaceutical characteristics of the resulting substance: manufacturing method defines conditions for the formation of highly bioavailable organometallic zinc nanoparticles. 

## Figures and Tables

**Figure 1 pharmaceutics-15-00590-f001:**
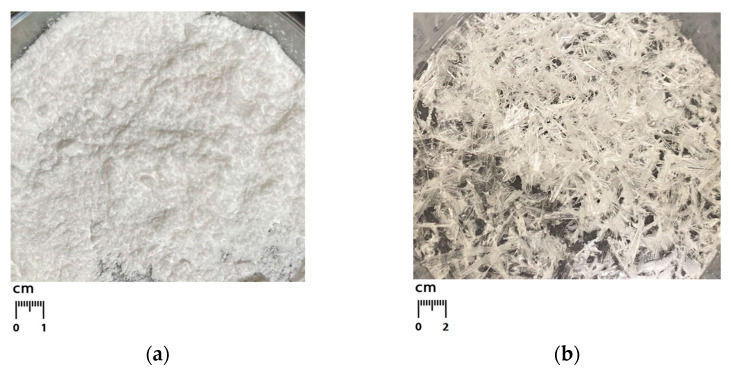
Appearance of synthesized zinc methionine Zn(Met)_2_ (**a**) and crystallized zinc methionine Zn(Met)_2_(SO_4_)_x_·nH_2_O (**b**) complexes.

**Figure 2 pharmaceutics-15-00590-f002:**
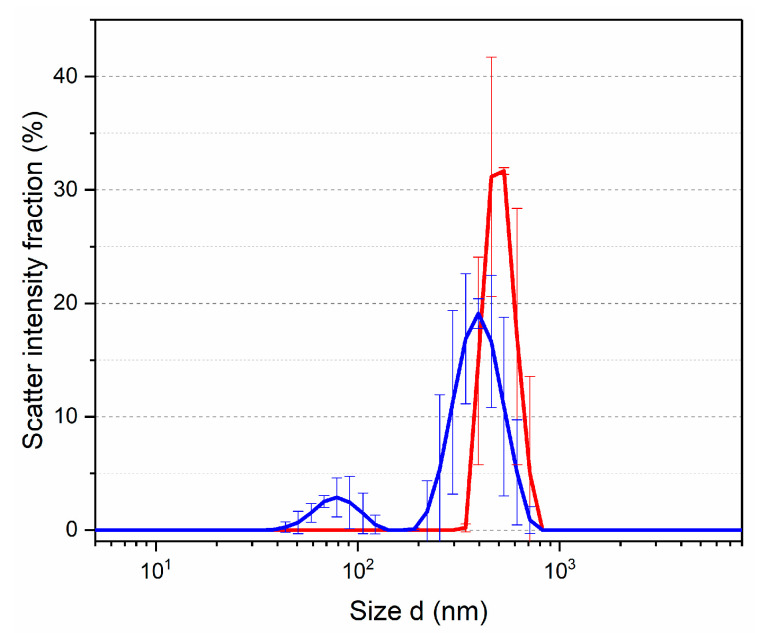
Particle size distribution in solutions of synthesized Zn(Met)_2_ (red) and crystallized Zn(Met)_2_(SO _4_)_x_·nH_2_O (blue) complexes (n = 3).

**Figure 3 pharmaceutics-15-00590-f003:**
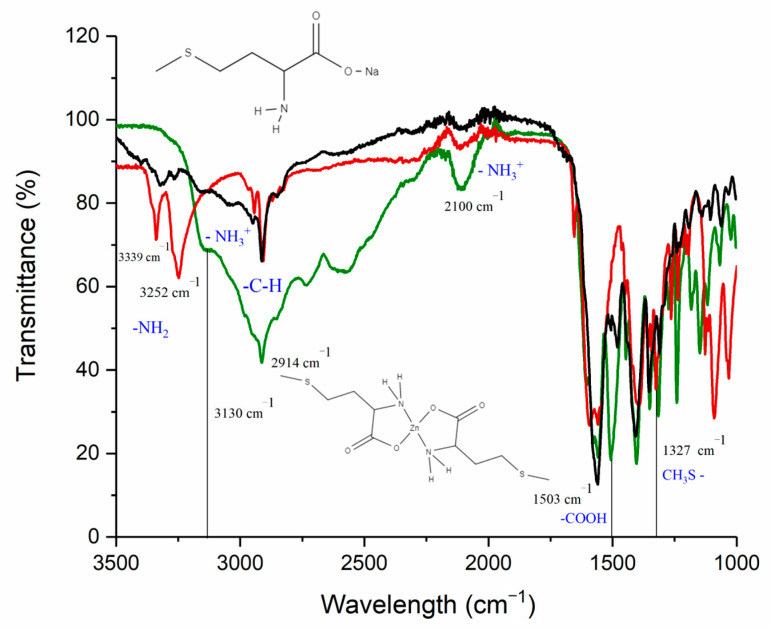
ATR-IR spectra of L-methionine (green), sodium methionine (black) and synthesized zinc methionine complex (red).

**Figure 4 pharmaceutics-15-00590-f004:**
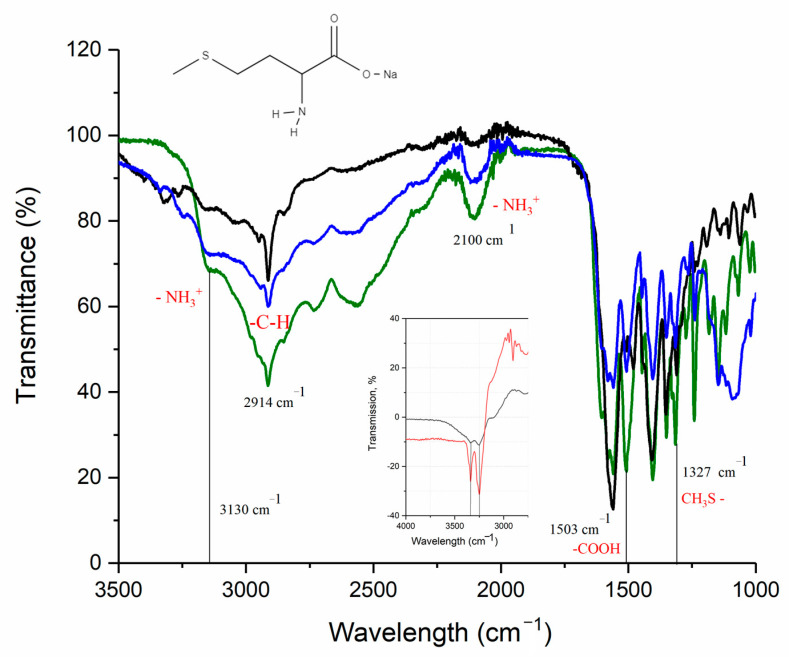
ATR-IR spectra of L-methionine (green), sodium methionine (black), and crystallized zinc methionine complex Zn(Met)_2_(SO_4_)_x_·nH_2_O (dark blue). Insert: difference of IR spectra of synthesized (red) and crystallized (black) complexes and methionine.

**Figure 5 pharmaceutics-15-00590-f005:**
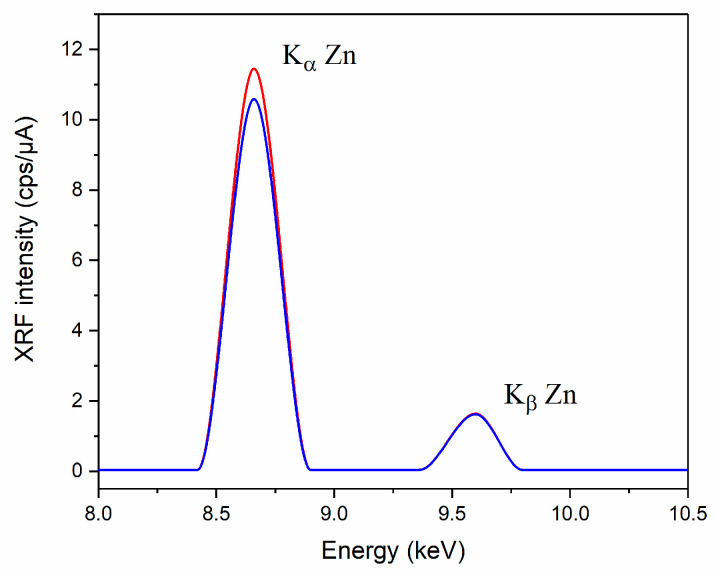
XRF spectra of synthesized Zn(Met)_2_ (red) and crystallized Zn(Met)_2_(SO_4_)_x_ complexes (blue) at the characteristic energy of zinc fluorescence—zinc K_α_ line of 8.632 keV and zinc K_β_ line of 9.572 keV.

**Figure 6 pharmaceutics-15-00590-f006:**
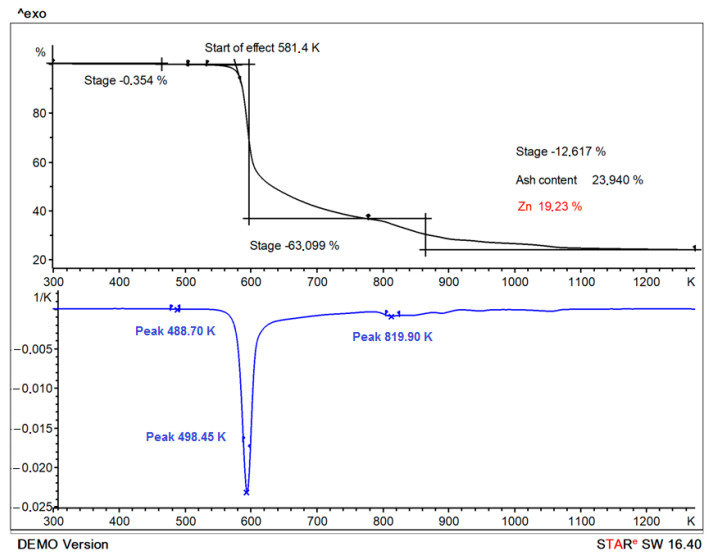
TGA and DTA curves of the synthesized Zn(Met)_2_ complex obtained by heating the sample in a static atmosphere of air in the temperature range from 298 to 1273 K.

**Figure 7 pharmaceutics-15-00590-f007:**
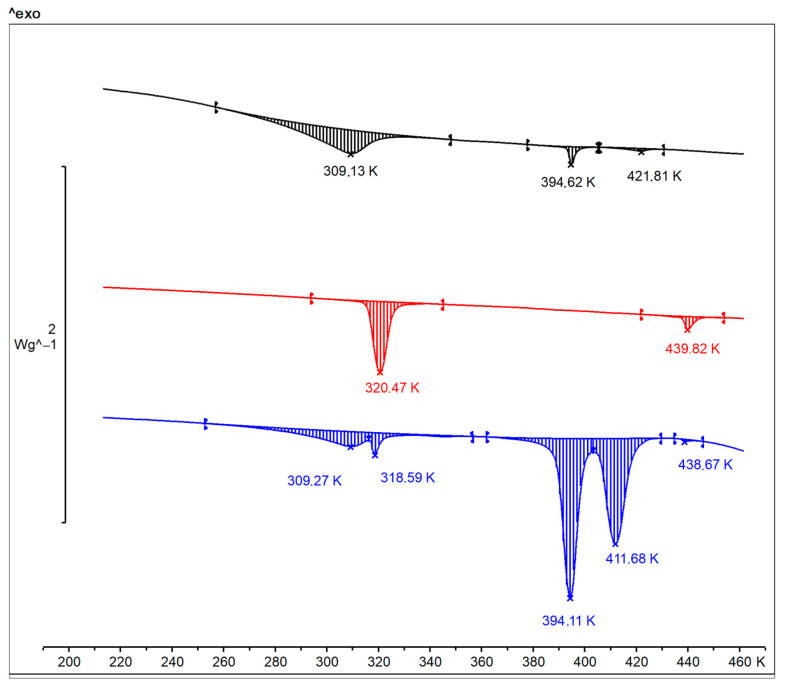
DSC thermogram of L-Methionine (black), synthesized Zn(Met)_2_ (red) and crystallized Zn(Met)_2_(SO_4_)_x_ (blue) complexes during the first heating of samples.

**Figure 8 pharmaceutics-15-00590-f008:**
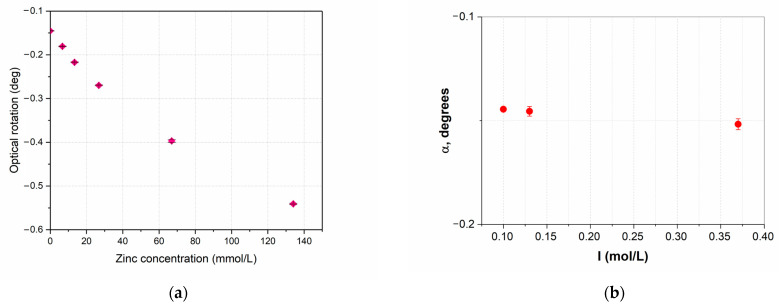
Optical activity of methionine and zinc sulfate solutions during the complexation reaction at pH 5.74 under conditions of increasing concentration of the complexing agent. The error bars represent the standard deviation of measurements (n = 5) (**a**). Control of ionic strength influence on the optical activity of methionine solution (134 mmol/L) without added zinc (**b**).

**Figure 9 pharmaceutics-15-00590-f009:**
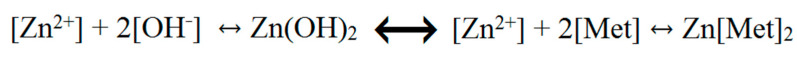
The competitive processes in aqueous solutions containing Zn^2+^ and Met in different molar ratios, pH = pI_met_.

**Figure 10 pharmaceutics-15-00590-f010:**
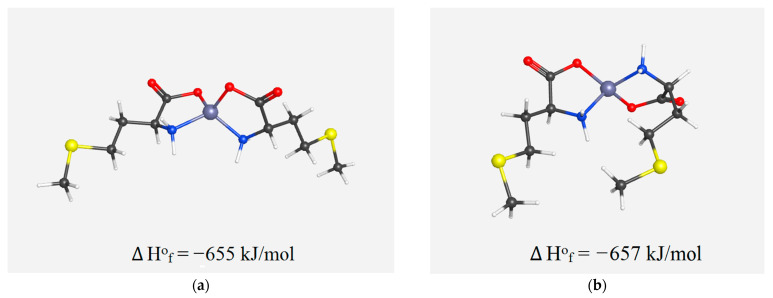
Spatial structure of chelate complexes cis-Zn(Met)_2_ (**a**) and trans-Zn(Met)_2_ (**b**) with calculated values of heat of formation ∆H°_f_.

**Figure 11 pharmaceutics-15-00590-f011:**
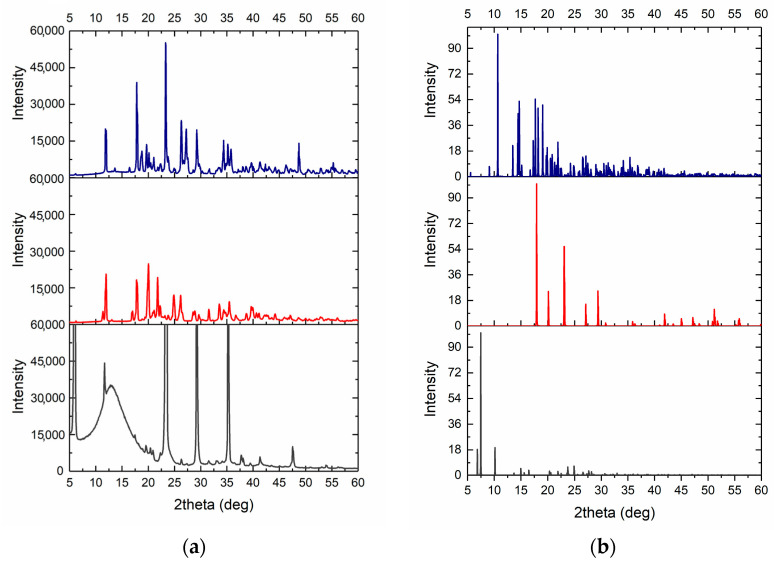
**Obtained** XRD pattern of synthesized Zn(Met)_2_ (red) and crystallized Zn(Met)_2_(SO_4_)_x_ (blue) complexes and recrystallized L-Methionine (gray) (**a**) and corresponding simulated XRD patterns (**b**).

**Table 1 pharmaceutics-15-00590-t001:** Specific rotation of water solutions of Zn(Met)_2_ and L-Met at different pH, *w*/*v* concentration 2% (134 mmol/L).

pH	[α]^20^_D_ ± SD	
	Zn(Met)_2_	L-Met
−1.35	+14.00 ± 0.31	+23.24 ± 0.06
12.95	−2.64 ± 0.29	+2.19 ± 0.03

## Data Availability

Data supporting reported results can be found on request by e-mail alla.marukhlenko@yandex.ru.

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
