# Peer review of "Comparative Analysis of Physical and Chemical Properties of Differently Obtained Zn—Methionine Chelate with Proved Antibiofilm Properties (Part II)"

_pharmaceutics, 2023, doi:10.3390/pharmaceutics15020590_

Round 1

Reviewer 1 Report

The manuscript entitled "Comparative analysis of physical and chemical properties of differently obtained Zn - methionine metal-organic compounds with proved antibiofilm properties" represent a comparative analysis of the physical and chemical properties of Zn - methionine metal-organic compounds, which previously proved that they have antibiofilm properties.

Because the previously article has been published in the same journal, the authors can rename the present manuscript as part II.

Or the authors can change the final part of the title :"with proved antibiofilm properties"

In the abstract the phrase "The previously demonstrated activity of aqueous solutions of methionine and zinc salts against biofilms of uropathogenic bacteria prompted us to compare structure and properties of zinc methionine obtained by different methods [5]." can be removed from the title. It can be added another phrase which mentioned the potential applications of the prepared materials and the novelty of the study.

In which way was determined the appearance of the obtained chelated compounds?

It was stated that the obtained compounds may have higher pharmacokinetic parameters (absorption and distribution, compared with which compounds? Some reference data regrading similar compounds.

The gray line from Figure 3 is not visible. Maybe you can change it with another colour. The same for the chemical formula of compounds inside the Figure. Also, the same for Fig. 4.

Inset from Fig 8 is not visible.

The Conclusion part is too complex. Maybe some phases can be moved in the Discussion part, and the Conclusion part must be short and concise.

Author Response

  • Dear reviewer, we have corrected the title
  • We have removed reference [5] from the abstract
  • In which way was determined the appearance of the obtained chelated compounds? It was just a simple photo, made on smartphone camera with 2x optical zoom out.
  • Better water solubility of crystallized Zn-Met in compare with the synthetized substance just gave us a hint about the probable better pharmacokinetic parameters. As indicated in many papers Zn(Met)2, obtained by synthetic method, is practically insoluble in water. References were added.
  • Fig. 3, fig. 4 and fig. 8 were replaced
  • Conclusion was reduced and corrected     

Reviewer 2 Report

The manuscript entitled “Comparative analysis of physical and chemical properties of differently obtained Zn - methionine metal-organic compounds with proved antibiofilm properties” by Morozova et al describes the analysis results of zinc coordination complexes with methionine obtained by synthesis and simple crystallization from water solution. The obtained results underline the role of the synthesis route for the biopharmaceutical characteristics of the resulting substance. Therefore, the manuscript is marginally acceptable for publication in Pharmaceutics, while several issues should be carefully addressed.
1. The abstract section generally does not contain citations, so please delete the citation.
2. The single crystal structures of the Zn - methionine metal-organic compounds obtained by different methods (synthesis and simple crystallization from water solution) are very helpful to support the conclusions of this manuscript. Authors are requested to make their best efforts to obtain single crystal structure data.
3. A scale bar should be added to Figure 1 to show the sample grain sizes.
4. Figure 11 should be added to the simulated X-ray diffraction patterns for comparison.
5. Although the antibiofilm properties have been previously reported by the authors, the discussion addition of the antibiofilm properties of the Zn - methionine metal-organic compounds obtained by different synthetic methods in this manuscript will be an important contribution to the significance of the manuscript.

Author Response

Dear reviewer, we`ve made all corrections you were asking about:

1) The reference was removed from the abstract

 2) Dear reviewer, I`m afraid to disappoint your expectations, but unfortunately we are out of the substance right now. If it is not so critical, can we just focus on XRD and other obtained results?

3) A scale bar was added

4) simulated X-ray diffraction patterns added

5) Discussion of antibiofilm properties is added to the introduction

Reviewer 3 Report

The current work focuses on the Comparative analysis of physical and chemical properties of 2 differently obtained Zn - methionine metal-organic compounds 3 with proved antibiofilm properties. The experimental work appears to have been carried out well. However, a few points deserve attention for further publication. I suggest that it is accepted for publication after the following revisions:

- ABSTRACT: What parameters were optimized? Authors must include numbers with the results found. Stability of the Zn - methionine metal-organic compounds ? How much Zn - methionine metal-organic compounds were utilized to process? Furthermore, what are the conditions of reactions? Temperature, pH, ionic strength, for example. This information should be included in the abstract.

- INTRODUCTION:

- In this study,  Comparative analysis of physical and chemical properties of  differently obtained Zn - methionine metal-organic compounds with proved antibiofilm properties: physical and covalent were implied for Zn - methionine metal-organic compounds preparation? What the advantages? Additionally, the spacer arm, the steric hindrances for the Zn - methionine metal-organic compounds caused by this groups when compared to the others groups? These strategies used should be better explained in the manuscript.

- Zn - methionine metal-organic compounds are very special, having a peculiar mechanism of action. This information must be clear in the introduction to present manuscript.

- A paragraph describing the properties, application, mechanism of actuation to Zn - methionine metal-organic compounds must be included in the manuscript.

- What is the origin of this Zn - methionine metal-organic compounds? Is it a comercial? Modified? How was it produced?

- Zn - methionine metal-organic compounds: Improved catalytic, kinetic, and efficient increased the activity? This process needs to be explained in the introduction of the manuscript.

- Zn - methionine metal-organic compounds: What optimization strategy was used? Why was it used? This information needs to be explained in the introduction of the manuscript.

- Zn - methionine metal-organic compounds: The new material presented were compared with a commercial material? This information must be clear in the introduction.

- The Zn - methionine metal-organic compounds for its economic use is, therefore, essential since can improve the selectivity of the Zn - methionine metal-organic compounds repeatability, the range of substrates, and the separation of the Zn - methionine metal-organic compounds from the materials. Among other advantages of Zn - methionine metal-organic compounds, one can mention the increased Zn - methionine metal-organic compounds prevention and improvement in the stability of the three–dimensional structure. This information must be clear in the introduction.

- Another point that must be considered, in this study the authors is the inhibitory activity and specific affinity to Zn - methionine metal-organic compounds. This information must be clear in the introduction.

- In this study, Zn - methionine metal-organic compounds were purified? The conventional methods for purification are ultrafiltration, precipitation and affinity chromatography. However some of these methods are complicated, laborious, time-consuming and expensive. This information must be clear in the introduction.

- The contribution and importance of these studies in the work performed must be explained in the introduction of the manuscript.

MATERIALS:

- Include the concentration of solutions.

METHODS:

- Include the molar concentration of all the chemicals used, the way the methods are presented, not possible reproducibility.

- Preparation procedure: Please include more details, temperature, amount of Zn - methionine metal-organic compounds per gram, pH, molar ratio, ionic strength.

- RESULTS AND DISCUSSION:

- The influence of substrate systems to Zn - methionine metal-organic compounds was also investigated? The Enhanced stability of Zn - methionine metal-organic compounds showed how about stability ?

- The thermal stability Zn - methionine metal-organic compounds prepared is one of the most important application criteria for diferent applications. This stability depends on the Zn - methionine metal-organic compounds preparation strategy. It also depends on the stabilization of the Zn - methionine metal-organic compounds. This discussion could be improved. Please include in the manuscript.

- The stability in organic solvents, metal ions, or detergent enables its wide application in synthesis processes which nowadays are in great demand from the point of view of industrial. The effect of organic solvents on the Zn - methionine metal-organic compounds activity was studied? For example, in the presence of ethanol, methanol, dimethyl sulfoxide (DMSO), dioxane, n-hexane, tert-butanol, acetone or 2-propanol?

- Comparison of the Zn - methionine metal-organic compounds in terms of kinetic parameters: Authors need to compare these results with other results in the literature.

- Was determined the full loading of Zn - methionine metal-organic compounds prepared under the optimized conditions? This information must be clear in the manuscript.

- The Zn - methionine metal-organic compounds may experience aggregation (mainly near to the isoelectric point). This may be caused by undesired Zn - methionine metal-organic compounds - interactions where inactivation that can stabilize incorrect Zn - methionine metal-organic compounds structures. This results must be cleared.

- The optimization of Zn - methionine metal-organic compounds preparation process, the preparations shown having diffusion limitations? Considering the strategy presented in this manuscript. Please, this should be explained in the manuscript. What were the optimum conditions?

- Effect of solution pH since the solution pH affects the generation of hydroxyl radicals and also influences the surfasse charge and interface potential properties, it is one of the important factors. Zn - methionine metal-organic compounds showed considerable improves in the kinetic parameters in terms of activity, specific activity, Km and Vmax, optimum pH and Temperature?

- Reusability of Zn - methionine metal-organic compounds:  The reusability of Zn - methionine metal-organic compounds particles is very important while considering reactions. Reusability was accounted for continuous application of the Zn - methionine metal-organic compounds: Reusability studies showed that the remaining Zn - methionine metal-organic compounds assay was obtained to reduce with the increasing number of re-use cycles? The reusability of Zn - methionine metal-organic compounds without alteration in its load capacity of synthesis performance with the resulting biocatalyst, which is an advantage. After  cycles, please, an explanation of these results. What other factors can influence the results achieved? In addition, the results should be compared with other works of the literature in the same line of application.

- Enhanced stability of Zn - methionine metal-organic compounds:  Other factors that cause the loss of durability and stability of the biocatalysts should be explained in the manuscript.

- Please, check all references according to the author's instructions.

- Include more details in the figures (error bars) and tables captions.

- The manuscript must be formatted according to the journal's standards.

Author Response

Dear reviewer, we have made the demanded corrections:

1) Abstract - corrected

2) Introduction - corrected, even rewritten)

3) Materials and methods - corrected

4) Based on your comments and questions, we have revised the article quite a lot, especially in the introduction part. More fully revealed the meaning of the intended work. I assume that in this way of presenting the material, many of your questions will disappear by themselves, since they may have arisen due to an introduction that misled you about the work. Thank you very much for your comments!

Round 2

Reviewer 1 Report

The authors made all the necessary corrections requested by the reviewer.

There are some mirror errors:

- Added the appearance analysis and the used technique in the experimental part. "The appearance of the studied compounds was determined using...."

The following reference was added in the manuscript.[M.A. 327
Mamun, Omar Ahmed, P.K. Bakshi, M.Q. Ehsan. Synthesis and spectroscopic, magnetic 328
and cyclic voltammetric characterization of some metal complexes of methionine: 329
[(C5H10NO2S)2MII]; M II = Mn(II), Co(II), Ni(II), Cu(II), Zn(II), Cd(II) and Hg(II)// 330
Journal of Saudi Chemical Society (2010)14,2331], [Star L, van der Klis JD, Rapp C, Ward 331
TL. Bioavailability of organic and inorganic zinc sources in male broilers. Poult Sci. 2012 332
Dec;91(12):3115-20. doi: 10.3382/ps.2012-02314. PMID: 23155021.].

Indicate the number and renumbered the references in the text.

Author Response

Dear reviewer, the experimental part was fulfilled with requested date.

References - corrected and renumbered.